# Free Vibrations of Anisotropic Nano-Objects with Rounded or Sharp Corners

**DOI:** 10.3390/nano11071838

**Published:** 2021-07-15

**Authors:** Lucien Saviot

**Affiliations:** Laboratoire Interdisciplinaire Carnot de Bourgogne (ICB), UMR 6303 CNRS, Université Bourgogne Franche-Comté, 9 Av. A. Savary, BP 47870, 21078 Dijon, France; lucien.saviot@u-bourgogne.fr

**Keywords:** acoustic vibrations, nanoparticle, nanowire, Rayleigh–Ritz variational method

## Abstract

An extension of the Rayleigh–Ritz variational method to objects with superquadric and superellipsoid shapes and cylinders with cross-sections delimited by a superellipse is presented. It enables the quick calculation of the frequencies and displacements for shapes commonly observed in nano-objects. Original smooth shape variations between objects with plane, convex, and concave faces are presented. The validity of frequently used isotropic approximations for experimentally relevant vibrations is discussed. This extension is expected to facilitate the assignment of features observed with vibrational spectroscopies, in particular in the case of single-nanoparticle measurements.

## 1. Introduction

The normal modes of the free vibration of 3D objects are involved in a variety of different processes at the nanoscale through electron–vibration coupling. Vibrational spectroscopies take advantage of this to provide a means to study the nano-objects themselves through their vibrations and their electronic properties. Inelastic light scattering (low-frequency Raman or Brillouin) and time-resolved transient absorption measurements are such experimental approaches that have been extensively used in particular to study metallic nano-objects [1,2,3]. Models to describe the vibrations are required to interpret experimental measurements. Models based on continuum elasticity have been shown to be suitable even in the case of small nano-objects for which the number of atoms near the surface is large [4,5]. Analytical solutions exist for isotropic spheres and isotropic circular infinite cylinders. A few other analytical solutions exist in other cases, but only for some particular vibrations or with additional approximations [6,7]. Numerical methods have fewer restrictions. The finite element method (FEM) can handle objects with complex shapes and anisotropic elasticity. The Rayleigh–Ritz variational approach (RR) is an intermediate solution. It can handle some nonspherical shapes and anisotropic elasticity and be much faster than FEM in some cases. For this reason, it is a method of choice in resonant ultrasound spectroscopy (RUS) [8]. RUS is a method based on the measurement of the frequencies of the free vibrations of a solid object in order to determine the elastic tensor of the material of which the object is made. It has been successfully applied to macroscopic objects in materials science and geology [9]. Three-dimensional objects are prepared with known dimensions and shapes before the eigenfrequencies are measured. Their mass density is measured as well, leaving only the elastic tensor as unknown. The vibrations are calculated, and a numerical procedure is used to minimize the difference with the measured frequencies. Quick calculations are needed for this method to be convenient. The RR approach meets this requirement when integrals over the volume of the nano-object can be computed quickly, which is the case for some simple analytic shapes (spheroids, rectangular parallelepipeds, elliptical cylinders, etc.) because the integrals can be expressed analytically [10].

Several issues appear when trying to use acoustic resonances to study nano-objects. The natural mechanical frequencies cannot be measured with piezoelectric transducers, and vibrational spectroscopies must be used. These spectroscopies provide only a few natural frequencies in the general case due to restrictive selection rules, while the RUS method is effective only when many frequencies are available. However, recent works have shown that these selection rules can be broken for large nanoparticles (NPs) or for NPs in close proximity with other NPs [11,12]. Another issue is that it is often impossible to control the shape of nano-objects accurately enough to match one of the simple shapes mentioned before. For example, corner rounding in NPs is commonly observed [13], and it is known to modify the properties of the nano-objects such as the electronic properties of metallic NPs [14]. It seems therefore necessary to take into account the exact shape to describe the vibrations as well. This work focuses on extending the RR approach to rounded and spiky shapes with concave or convex faces and using it to study the influence of a few original shape variations on the frequencies.

## 2. Superquadrics and Superellipsoids

RR calculations were performed in this work using the xyz algorithm introduced by Visscher et al. [10]. The displacements are decomposed on the basis of functions $\phi_{\lambda }=x^{p}y^{q}z^{r}$ with *p*, *q*, and *r* integers and p+q+r≤N. The RR approach turns the dynamic problem into the generalized eigenvalue problem $\omega^{2}Ea=\Gamma a$ with ω the pulsation, *a* the amplitudes of the $\phi_{\lambda }$ components, and *E* and Γ two square matrices, defined as follows: (1)$$\begin{array}{ccc} E_{\lambda i\lambda^{\prime }i^{\prime }} & =\delta_{ii^{\prime }} & \int_{V}\phi_{\lambda }\rho \phi_{\lambda^{\prime }}dV \end{array}$$(2)$$\begin{array}{ccc} \Gamma_{\lambda i\lambda^{\prime }j} & =C_{iji^{\prime }j^{\prime }} & \int_{V}\phi_{\lambda ,j}\phi_{\lambda^{\prime },j^{\prime }}dV \end{array}$$

$C_{ijkl}$ is the stiffness tensor, ρ the mass density, and *V* the volume of the object. The xyz algorithm is very efficient when analytic expressions for the integrals exist. This is the case for spheroids, rectangular parallelepipeds, elliptical cylinders, and a few other shapes, as pointed out in the original work [10]. In addition, the Γ matrix is block-diagonal when the shape and elastic properties are symmetric with respect to the x=0, y=0, or z=0 planes [15]. In those cases, the linear system can be turned into a few smaller linear systems requiring less computer time and resources.

To extend the xyz algorithm to more shapes, analytic expressions for the volume integrals in the case of superquadrics and superellipsoids are derived in the following. The surface of superquadrics and superellipsoids is given by the implicit Equations (Equation 3) and (4) where $L_{i}$ is the half-length in the *i* direction and $n_{i}>0$. The shape is an octahedron if $n_{i}=1,\forall i$ and spheroidal if $n_{i}=2,\forall i$, and it tends to a rectangular parallelepiped when $n_{i}\to \infty ,\forall i$. A large variety of shapes is obtained by independently varying the five or six parameters because superquadrics and superellipsoids are different except when $n_{x}=n_{y}=n_{xy}=n_{z}$. In particular, the shape can be varied continuously among cubes (or rectangular parallelepipeds), spheres (or spheroids), and octahedra and octahedra with concave faces, as illustrated in Figure 1, for $L=L_{i}$ and $n=n_{i},\forall i$. It is worth noting that similar NP shapes have been reported in the literature. Reports about spheres are numerous. There are many works on nanocubes, and it is worth noting that, most often, their corners are rounded even when they are considered to be sharp [13,16]. Such rounded nanocubes are usually modeled by imposing a radius of curvature at the edges. The superquadric shape makes it possible to control the rounding of the corners through the *n* shape parameter. One disadvantage of this approach is that the faces are not flat near their center. However, the deviation from a perfectly flat surface is very small for large enough *n*. Finally, the shape obtained for n<1 is reminiscent of nanostars [17]. The present approach is not flexible enough to accurately model the large variety of reported nanostars because the number and position of the branches are fixed and the tips are not rounded. Still, it provides a very quick method to approach such a complex geometry. To the best of my knowledge, using the xyz algorithm for such shapes has not been reported [10].
(3)$$\begin{array}{cc} {\left|\frac{x}{L_{x}}\right|}^{n_{x}}+{\left|\frac{y}{L_{y}}\right|}^{n_{y}}+{\left|\frac{z}{L_{z}}\right|}^{n_{z}} & =1 \end{array}$$
(4)$$\begin{array}{cc} {\left({\left|\frac{x}{L_{x}}\right|}^{n_{xy}}+{\left|\frac{y}{L_{y}}\right|}^{n_{xy}}\right)}^{\frac{n_{z}}{n_{xy}}}+{\left|\frac{z}{L_{z}}\right|}^{n_{z}} & =1 \end{array}$$

## 3. Volume Integrals

The expressions in Equations (Equation 1) and (2) come down to evaluating *f*, the volume integral of power functions $x^{p}y^{q}z^{r}$, over the volume *V* of the object (Equation (Equation 5)). In the following, only the octant with x≥0, y≥0, and z≥0 is considered. The integral in this octant is denoted $f_{8}$. The values in the seven other octants have the same absolute value, and the signs are deduced from the parity of *p*, *q*, and *r*.
(5)$$\begin{array}{cc} f(p,q,r) & =\int_{V}x^{p}y^{q}z^{r}dV\\ f_{8}\left(p,q,r\right) & =\int_{V\cap \{x,y,z\geq 0\}}x^{p}y^{q}z^{r}dV \end{array}$$

### 3.1. Superquadrics

In the octant mentioned before, the volume of the superquadrics can be described by the parametric description given in Equation (Equation 6). The intermediate variables (X,Y,Z) are introduced. Equation (Equation 3) becomes $X^{2}+Y^{2}+Z^{2}=1$, which allows further parametrizing by introducing the spherical-like variables r∈[0,1], $\theta \in [0,\frac{\pi }{2}]$, and $\phi \in [0,\frac{\pi }{2}]$.
(6)$$\left\{\begin{array}{cc} {\left(\frac{x}{L_{x}}\right)}^{\frac{n_{x}}{2}}=X & =r\sin\theta \cos\phi \\ {\left(\frac{y}{L_{y}}\right)}^{\frac{n_{y}}{2}}=Y & =r\sin\theta \sin\phi \\ {\left(\frac{z}{L_{z}}\right)}^{\frac{n_{z}}{2}}=Z & =r\cos\theta \end{array}\right.$$

After having performed these changes of variables, the volume integral is the product of three one-dimensional integrals over *r*, θ, and ϕ. The ones over θ and ϕ involve products on sine and cosine functions with various powers, which can be expressed with the *B* or Γ functions using Equation (Equation 7) ([18], Equations 5.12.1 and 5.12.2). The calculation steps are given in Appendix A, and the resulting expression for $f_{8}$ is given in Equation (Equation 8).
(7)$$B\left(a,b\right)=\frac{\Gamma (a)\Gamma (b)}{\Gamma (a+b)}=2\int_{0}^{\frac{\pi }{2}}\sin^{2a-1}t\cos^{2b-1}tdt$$
(8)$$f_{8}\left(p,q,r\right)=\frac{L_{x}^{p+1}L_{y}^{q+1}L_{z}^{r+1}}{\left(p+1)(q+1)(r+1\right)}\frac{\Gamma \left(\frac{p+1}{n_{x}}+1\right)\Gamma \left(\frac{q+1}{n_{y}}+1\right)\Gamma \left(\frac{r+1}{n_{z}}+1\right)}{\Gamma \left(\frac{p+1}{n_{x}}+\frac{q+1}{n_{y}}+\frac{r+1}{n_{z}}+1\right)}$$

### 3.2. Superellipsoids

Similarly, using the parametric representation for superellipsoids given in Equation (Equation 9), which is valid in the same octant, the volume integral can be transformed into the product of three integrals and expressed as a function of the *B* or Γ functions. The calculation steps are given in Appendix B, and the resulting expression is given in Equation (Equation 10).
(9)$$\left\{\begin{array}{cc} x & =rL_{x}\sin^{\frac{2}{n_{z}}}\theta \cos^{\frac{2}{n_{xy}}}\phi \\ y & =rL_{y}\sin^{\frac{2}{n_{z}}}\theta \sin^{\frac{2}{n_{xy}}}\phi \\ z & =rL_{z}\cos^{\frac{2}{n_{z}}}\theta \end{array}\right.$$
(10)$$f_{8}\left(p,q,r\right)=\frac{L_{x}^{p+1}L_{y}^{q+1}L_{z}^{r+1}}{\left(p+1)(q+1)(r+1\right)}\frac{\Gamma \left(\frac{p+1}{n_{xy}}+1\right)\Gamma \left(\frac{q+1}{n_{xy}}+1\right)\Gamma \left(\frac{r+1}{n_{z}}+1\right)}{\Gamma \left(\frac{p+q+r+3}{n_{z}}+1\right)}\frac{\Gamma \left(\frac{p+q+2}{n_{z}}+1\right)}{\Gamma \left(\frac{p+q+2}{n_{xy}}+1\right)}$$

### 3.3. Nanowires

A similar approach can be used to model acoustic phonons in nanowires (NWs) [3,19,20]. In that case, the integrals of interest are over the cross-section *S* instead of the volume, and calculating the elements of the *E* and Γ matrices comes down to calculating *f* as defined in Equation (Equation 11). As before, we consider only the x≥0 and y≥0 quadrant, and the value of the integral in this domain is noted $f_{4}$. The integrals in the three other quadrants are obtained by symmetry from the parity of *p* and *q*.
(11)$$\begin{array}{cc} f(p,q) & =\int_{S}x^{p}y^{q}dS\\ f_{4}\left(p,q\right) & =\int_{S\cap \{x,y\geq 0\}}x^{p}y^{q}dS \end{array}$$

Let us consider an NW aligned along *z* having a cross-section delimited by a superellipse defined by implicit Equation (Equation 12).
(12)$${\left|\frac{x}{L_{x}}\right|}^{n_{x}}+{\left|\frac{y}{L_{y}}\right|}^{n_{y}}=1$$

The analytic expression for $f_{4}$ obtained using the same approach as before is given in Equation (Equation 13).
(13)$$f_{4}\left(p,q\right)=\frac{L_{x}^{p+1}L_{y}^{q+1}}{\left(p+1)(q+1\right)}\frac{\Gamma \left(\frac{p+1}{n_{x}}+1\right)\Gamma \left(\frac{q+1}{n_{y}}+1\right)}{\Gamma \left(\frac{p+1}{n_{x}}+\frac{q+1}{n_{y}}+1\right)}$$

### 3.4. Comparison with Previous Works

The analytic formula for spheroids given by Visscher et al. [10] can be obtained from Equation (Equation 8) or (Equation 10) by setting $n_{i}=2,\forall i$. The expression for the rectangular parallelepiped is recovered with $\frac{1}{n_{i}}=0$. Similarly, finite circular cylinders aligned along *z* with flat ends correspond to superquadrics or superellipsoids with $\frac{1}{n_{z}}=0$ and $n_{i\neq z}=2$. As expected, analytic expressions for $f_{8}$ equivalent to those in Visscher et al. [10] were obtained using these values. The expressions for infinite circular and square cylinders also match those obtained in previous works [19,20].

## 4. Results

### 4.1. General Case

NPs of low symmetry are easily obtained by using superquadrics with different values for $L_{i}$ and $n_{i}$ (i∈{x,y,z}) in each octant. The surface of the resulting shape is not smooth as it includes parts of the x=0, y=0, and z=0 planes. More interestingly, the surfaces of the eight superquadrics are connected across the x=0, y=0, and z=0 planes when using two sets of parameters for positive and negative values of *i*: $\left(L_{x}^{+},n_{x}^{+}\right)$, $\left(L_{x}^{-},n_{x}^{-}\right)$, *…*$\left(L_{z}^{-},n_{z}^{-}\right)$. This shape fits inside the $\left(L_{x}^{+}+L_{x}^{-}\right)\times \left(L_{y}^{+}+L_{y}^{-}\right)\times \left(L_{z}^{+}+L_{z}^{-}\right)$ rectangular parallelepiped. Its length along *i* is exactly $\left(L_{i}^{+}+L_{i}^{-}\right)$. The resulting shapes are still quite complex because different parts of the surface can be concave or convex. In addition, the surface is of class $C^{k}$ if $n_{i}^{\pm }>k,\forall i$. In the general case, such shapes are only invariant under the identity operation. In the following, we consider symmetric shapes that are at least invariant by reflection through the x=0, y=0, and z=0 planes by choosing $L_{i}=L_{i}^{+}=L_{i}^{-}$ and $n_{i}=n_{i}^{+}=n_{i}^{-},\forall i$. If the elastic tensor is also invariant under the same reflections, the linear system is block-diagonal and can be solved for each irreducible representation separately within a few seconds with modern processors [10,15]. In this work, calculations were performed using the eigensystems and special functions provided by the GNU Scientific Library [21]. The method is stable up to N=20, which was used throughout this work. For an actual implementation running in a web browser, see [22].

When the same set of parameters is used in the eight octants and for the three directions ($L=L_{x}=L_{y}=L_{z}$ and $n=n_{x}=n_{y}=n_{z}$), the shape of the NPs has cubic symmetry ($O_{h}$ point group). The point group is unchanged if the elasticity of the material the NP is made of is cubic with the lattice directions corresponding to *x*, *y*, and *z* or better (isotropic). Similarly, NWs and nanorods (NRs) with identical parameters along *x* and *y* have tetragonal symmetry (D_4h_), provided the elasticity is tetragonal or better (cubic or isotropic) and the lattice is aligned along *x*, *y*, and *z* as well.

In the following, such cubic NPs or tetragonal NWs and NRs made of cubic gold are considered. We focused on modes coming from the Raman active vibrations of the isotropic sphere, which are the experimentally relevant ones [23]. In particular, we checked the validity of common isotropic approximations to predict the frequencies of such vibrations. Original continuous shape variations among objects with plane, concave, and convex faces were investigated.

In the framework on continuum elasticity, the eigenfrequencies scale as the inverse of a characteristic length. In other words, if the lengths of a given nano-object are all multiplied by α, then all its eigenfrequencies are divided by α. For a sphere, the frequencies ν vary as 1/radius. For a cube, the frequencies vary as 1/edgelength. In other words, ν×radius or ν×edgelength is a constant for a given mode. In order to define a characteristic length appropriate for objects having different shapes, it is useful to remember that frequency is related to mass. For this reason, in the following, we consider the characteristic length defined as the cube root of the volume (*V*) for finite NPs or the square root of the surface area of the cross-section (*S*) for infinite cylinders. The product of the frequency with one of these characteristic lengths is considered in the following. It is expressed in m/s.

### 4.2. Breathing Modes

Breathing modes are precisely defined for spheres made of an isotropic material. In that case, they correspond exclusively to spheroidal vibrations with angular momentum ℓ=0, which are the only vibrations for which the volume changes during oscillation. For nonspherical or anisotropic NPs, no such simple assignment exists [24]. Volume varies for totally symmetric vibrations only (i.e., vibrations for which the deformed shape is invariant for all the symmetry operations), but the variations are very different and can be very small for some of these vibrations, contrary to the previous case. To identify breathing-like modes in the following, we focused on the ones with the largest volume variation. We considered NPs made of gold (ρ=19.293 g/cm^3^) with an isotropic approximation ($C_{11}=213.83$ and $C_{12}=153.57$ GPa) or cubic elasticity ($C_{11}=191$, $C_{12}=162$, and $C_{44}=42.4$ GPa).

Figure 2 (left) shows the frequency variations of the A_1g_ vibrations for isotropic gold. The shapes correspond to those plotted in Figure 1. The frequencies were normalized by multiplying by the cubic root of the volume *V* of the NPs. Note that this normalization plays a very significant role because the volume was multiplied by 90 when *n* varies from ½ to *∞*, while *L* was kept constant. The thickness of the lines is proportional to the relative volume variation during oscillation ΔV/V. ΔV is the volume integral of the divergence of the normalized displacement. Since the displacements are expanded on an $x^{l}y^{m}z^{n}$ basis, it was calculated using the expressions derived in the previous section. As written above, pure breathing modes exist only for the sphere. Indeed, the thickness of all the branches vanishes at n=2 except for the spheroidal modes with ℓ=0. Two horizontal red lines mark the frequencies of the fundamental mode and the first overtone. For nonspherical shapes, the largest volume variation occurs close to these two lines. Assuming the existence of a breathing mode at the frequency of a sphere having an identical volume is therefore a good approximation. For octahedra with concave faces (n<1), this approximation looks less and less valid as *n* decreases. In particular, a deviation toward larger frequencies is observed for the first overtone. One reason for this deviation could be the poor convergence of the xyz algorithm. Indeed, as *n* decreases, the branches of the nanostars become thinner and thinner. This manifests in the calculations with large displacements at the tips, which might be an indication that the convergence is not good enough. However, the frequency range where the volume variation is large is obtained in the other cases (n>1) even with quite small values of *N* (N<10). Therefore, it is reasonable to assume that the same pattern holds for 1/2≤n≤1 with N=20 and that the observed deviation is real.

Because nanocrystals (NCs) are seldom elastically isotropic, we considered next the case of NPs made of cubic gold. The cubic lattice is aligned with the *x*, *y*, and *z* axes to preserve the cubic symmetry (O_h_). Figure 2 (right) shows the frequency variation of all the A_1g_ vibrations. As in the previous figure, the frequencies of the first two breathing modes of a sphere made of the isotropic approximation of gold are plotted as horizontal lines. Complex patterns between the A_1g_ branches are observed in Figure 2 including in the case of the sphere. Still, the isotropic approximation is also valid with the same restriction as in the case of isotropic elasticity.

Using a similar approach, the radial breathing modes of infinite cylinders made of isotropic (left) and cubic (right) gold are presented in Figure 3 as a function of the shape of the cross-section. The frequencies are normalized by multiplication with the square root of the cross-section surface area *S*. The width of the lines is proportional to the surface variation ΔS/S. The modes of interest are the A_1g_ modes at the center of the Brillouin zone (q=0). In this case, the horizontal lines correspond to the first two radial breathing mode frequencies of the circular isotropic NW. The same observations as before apply. The circular isotropic approximation is quite good for n>1. For n<1, deviations toward higher frequencies appear reaching ∼+15% for the fundamental mode and ∼+10% for the first overtone at n=1/2.

### 4.3. Quadrupolar Modes

The quadrupolar modes of spherical NPs (spheroidal modes with ℓ=2, degeneracy 2ℓ+1=5) are of interest as well, in particular because they play a significant role in inelastic light-scattering experiments. In cubic symmetry, they split into the E_g_ (degeneracy 2) and T_2g_ (degeneracy 3) irreducible representations [25] making it a very useful signature of elastic anisotropy in spherical NPs, in particular for strongly anisotropic materials such as gold [26]. Figure 4 shows the variations of the normalized frequencies for the E_g_ (left) and T_2g_ (right) vibrations when varying the superquadric shape. Only cubic elasticity is considered in the following as in Figure 2 (right). The lowest frequency E**g** mode agrees quite well with the isotropic spherical approximation prediction for a sphere having the same volume over the full *n* range. This approximation is obtained using the transverse speed of sound of gold in the [110] direction [27]. A similar approximation exists for the T_2g_ mode using the [100] direction. It works reasonably well also, in particular, for n>1. For n<1, there are many anticrossing patterns with branches whose frequency decreases quickly with decreasing *n* (bending of the branches of the “nanostars”).

Similarly, the frequencies of the quadrupolar-like vibrations of NWs when varying the shape of the cross-section are plotted in Figure 5. As discussed in a previous work [20], these are the B_1g_ and B_2g_ modes (D_4h_ point group). A similar result was obtained, namely a very good agreement with the circular isotropic approximation for the lowest B_1g_ frequency and a mode complex picture for the B_2g_ vibrations with anticrossing patterns for n<1, but otherwise a good agreement as well. As before, different transverse sound speeds were used for both modes. Note that for n=1, the shape of the cross-section is a square rotated by 45 degrees around the *z* axis, i.e., a square with faces aligned with the [110] direction. As expected, even if the shapes are identical for n=1 and n→∞, the frequencies differ because of the different orientations of the lattice structure.

A very similar picture is obtained in both cases (NPs and NWs) for isotropic gold (not shown). In that case, anisotropy comes from the shape only. For the NPs, the frequencies of the spherical isotropic approximations are the same for E_g_ and T_2g_ (nondegenerate spheroidal mode with ℓ=2). For the NWs, the frequencies of the circular isotropic approximations are also the same for B_1g_ and B_2g_ (nondegenerate |m|=2 mode).

### 4.4. Edge Modes in Nanorods

In a previous work [20], it was shown that eigenmodes in finite-length circular cylinders (NRs) can be approximated by standing waves of the corresponding infinite cylinder or NW. The resulting mode frequencies fall therefore in the frequency range of the corresponding phonon branch. Additional modes having the same symmetry have been reported. They are localized at both ends of the finite cylinder. Their frequency can be lower than the minimum frequency of the branch, and it reaches a minimum when the ends of the NRs are flat. This behavior can be tested easily with the superquadratic and superellipsoid shapes instead of half spheroids positioned at the ends of the NRs, as in the previous work. Figure 6 presents the B_1g_ (bottom) and B_2g_ (top) frequencies for NRs (left, $L_{z}/L_{xy}=3$) and NWs (right) having a circular cross-section. In both cases, frequencies below the corresponding phonon band minimum are observed for large $n_{z}$, showing the presence of edge modes for NRs when their ends are flat or almost flat. These edge modes appear at lower $n_{z}$ for superellipsoids. This is because for a given $n_{z}$, superellipsoids are closer to rods with flat ends than superquadrics.

## 5. Conclusions

An extension of the xyz algorithm to NPs having superquadratic and superellipsoid shapes and NWs having a superellipse cross-section was proposed. It enables quickly assessing the vibrational properties of commonly reported nano-objects such as rounded nanocubes and nanostars. For nano-objects having the *x*, *y*, or *z* planes as mirror planes, the method enables quickly calculating the eigenmodes having a specific irreducible representation. The possibility to smoothly vary the shape of NPs from a cube to a sphere, an octahedron, and then an octahedron with concave faces was used to examine the evolution with the shape of experimentally relevant vibrations. The frequencies of the breathing and quadrupolar vibrations can be estimated reliably in most cases from the corresponding frequency for an isotropic sphere having the same volume or a circular isotropic cylinder having the same cross-section surface area. For the quadrupolar vibrations, the anisotropy of the material the nano-objects are made of must be taken into account for this approximation to be useful. The diverse set of shapes offered by this extension to the xyz algorithm makes it easier to model the exact shape of actual NPs. This is of interest in the context of single-NP measurements. In that case, the experimental features are narrow, and they depend on the actual shape of the NPs, which is in general neither perfectly rounded, nor having perfectly flat faces.

## Figures and Tables

**Figure 1 nanomaterials-11-01838-f001:**
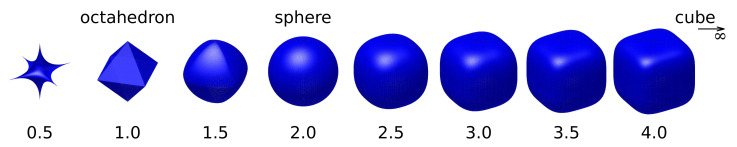
Superquadrics or superellipsoids with $L_{x}=L_{y}=L_{z}$. The value of $n=n_{x}=n_{y}=n_{z}=n_{xy}$ is shown below each shape.

**Figure 2 nanomaterials-11-01838-f002:**
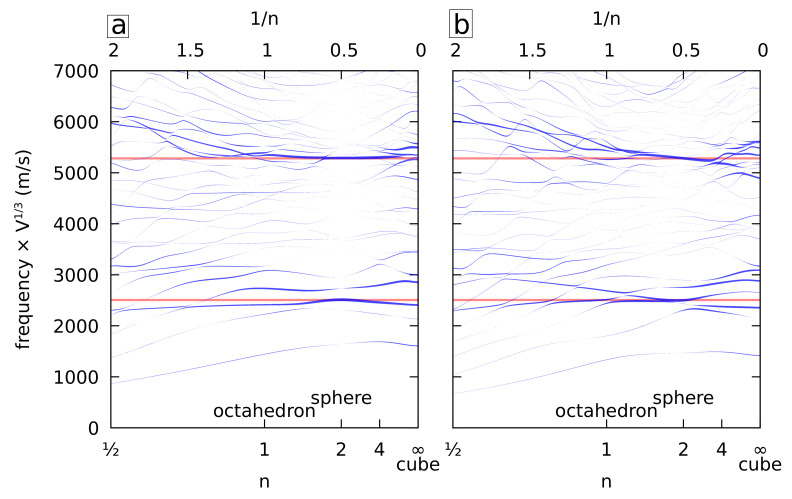
Normalized frequencies of the A_1g_ vibrations (O_h_) of isotropic (**a**) and cubic (**b**) gold superquadrics (or superellipsoids) with identical dimensions ($L_{i}$) and shape factors ($n_{i}=n$) along *x*, *y*, and *z* as a function of *n*. The frequencies are multiplied by the cubic root of the volume. The width of the lines is proportional to the relative volume variation of each mode ΔV/V. The horizontal red lines correspond to the first two breathing mode frequencies of the isotropic gold sphere.

**Figure 3 nanomaterials-11-01838-f003:**
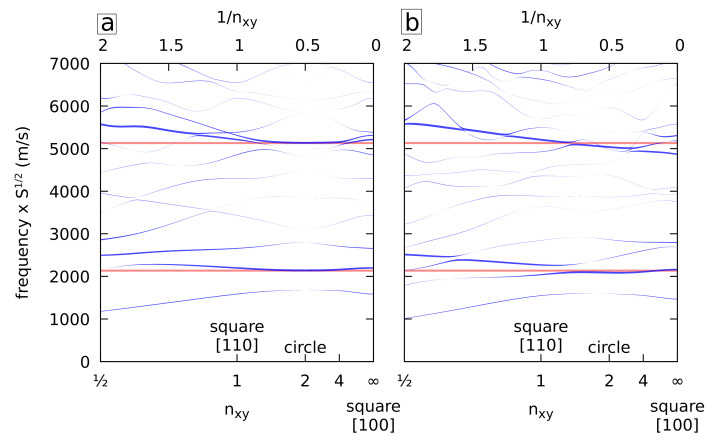
Normalized frequencies of the A_1g_ vibrations (D_4h_) at the center of the Brillouin zone (q=0) for isotropic (**a**) and cubic (**b**) gold NWs with cross-sections delimited by superellipses with $L_{x}=L_{y}$ as a function of $n_{x}=n_{y}=n_{xy}$. The frequencies are multiplied by the square root of the surface. The width of the lines is proportional to the relative volume variation of each mode ΔS/S. The horizontal red lines correspond to the first two radial breathing mode frequencies of the isotropic circular gold NW.

**Figure 4 nanomaterials-11-01838-f004:**
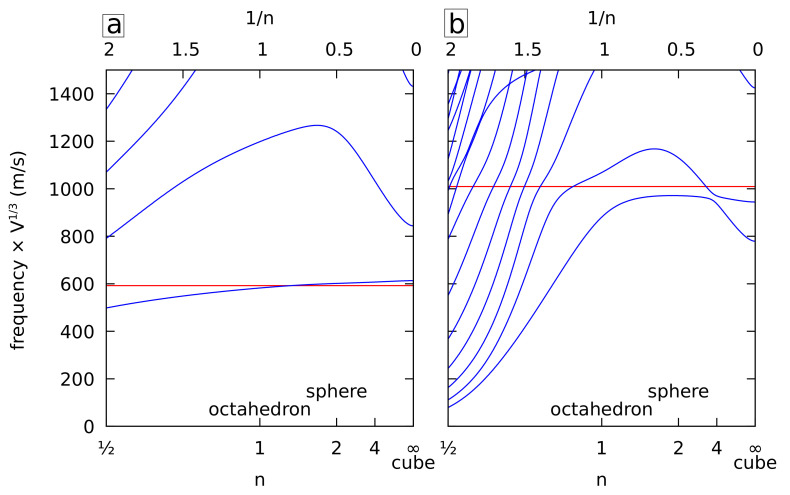
Normalized frequencies of the E_g_ (**a**) and T_2g_ (**b**) vibrations of cubic gold superquadrics (or superellipsoids) with identical dimensions ($L_{i}$) and shape factors ($n_{i}=n$) along *x*, *y*, and *z* as a function of *n*. The frequencies are multiplied by the cubic root of the volume. The lowest horizontal red line corresponds to the frequency of an isotropic sphere with the longitudinal and transverse sound velocities of cubic gold in the 110 (**a**) and 100 (**b**) directions.

**Figure 5 nanomaterials-11-01838-f005:**
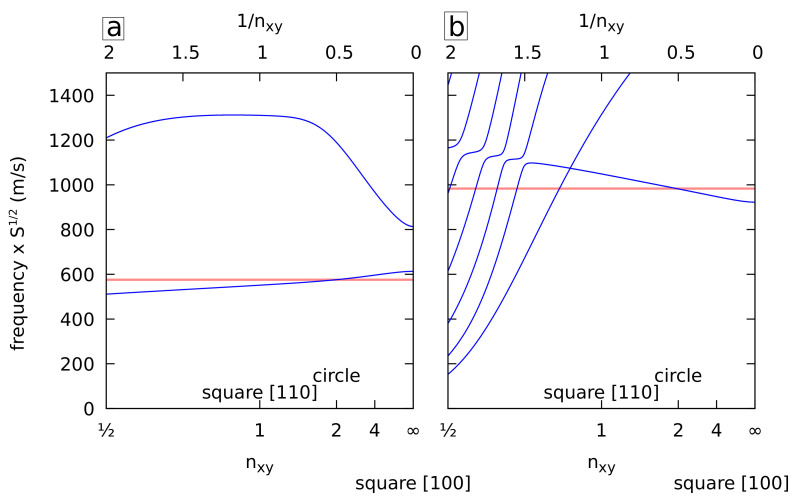
Normalized frequencies of the B_1g_ (**a**) and B_2g_ (**b**) vibrations of cubic gold cylinders with cross-sections delimited by superellipses with $L_{x}=L_{y}$ as a function of $n_{x}=n_{y}=n_{xy}$. The frequencies are multiplied by the square root of the surface area. The lowest horizontal red line corresponds to the frequency of an isotropic cylinder with the longitudinal and transverse sound velocities of cubic gold in the 110 (**a**) and 100 (**b**) directions.

**Figure 6 nanomaterials-11-01838-f006:**
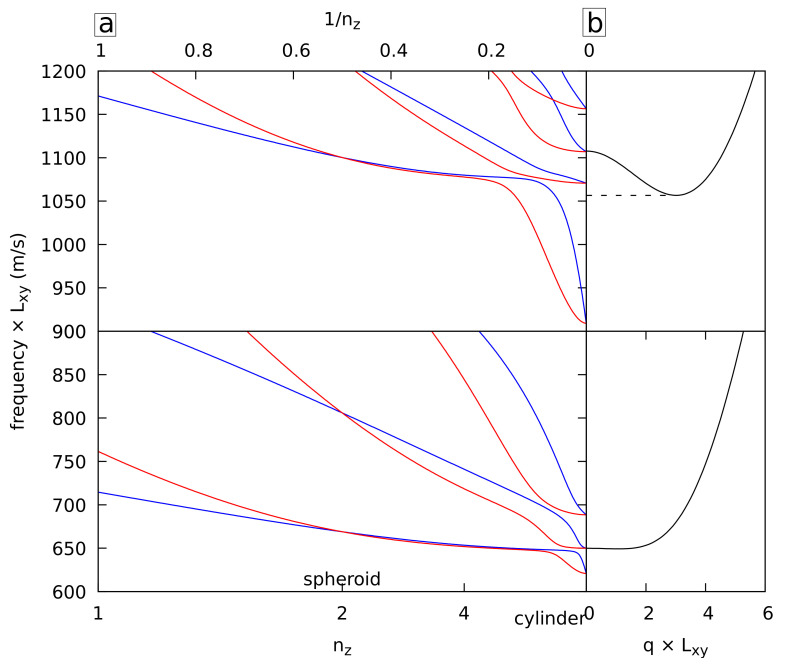
(**a**) Normalized frequencies of the B_1g_ vibrations of cubic gold superquadrics (blue) and superellipsoids (red) with $L_{z}/L_{xy}=3$ and $n_{x}=n_{y}=n_{xy}=2$ as a function of $n_{z}$. The dashed horizontal line corresponds to the minimum frequency of the corresponding phonon branch of the circular NW. (**b**) Corresponding phonon branches of the NW.

## Data Availability

The datasets used and/or analyzed during the current study are available from the corresponding author on reasonable request.

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
