# Peer review of "Free Vibrations of Anisotropic Nano-Objects with Rounded or Sharp Corners"

_nanomaterials, 2021, doi:10.3390/nano11071838_

Round 1

Reviewer 1 Report

well written paper, where the author has already dealt with issues on the same subject in other articles.
It would be appropriate to expand the bibliography.
Discuss the possible implications and practical applications of this study

Author Response

Thank you for the suggestion. The bibliography was expanded in two areas. One reference was added to a review about acoustic phonons in nanowires (new reference 3). Two other references were added regarding using a continuum approximation for nanoparticles (new references 4 and 5). I believe these references highlight the relevance of the present work to the field.

Reviewer 2 Report

The vibrational spectroscopy has the fundamental importance in understanding the physical properties of nanomaterials. This manuscript,  which extends the Rayleigh-Ritz variational method to objects with superquadric and superellipsoid shapes, is interesting and well written. It contains a novel ideas to calculate the vibrational frequency for nano particles with various shape.

The referee suggests the author adds some discussions on a special issue, namely how characteristic vibrational frequencies, e.g, breathing modes, change with the size of nano particles. After all, for people who study nano materials, they are more interested in how the physical properties changing with the size of systems.

Author Response

Thank you for the comments. A paragraph was added at the end of section 4.1 to discuss the size variation and how to define a relevant characteristic length for objects having different shapes. The previous version of the manuscript was indeed lacking in this domain. Thank you for the suggestion.

Reviewer 3 Report

Authors claim that the nano-objects with sharp or rounded corners are dynamically analyzed. However, no one can see how the nanoscale effects are imposed on the mathematical formulations. For example, surface effects, nonlocal impact, strain gradient effect, etc. These effects are so crucial in nanoscale in particular for such studied nanoparticles. The authors must explain this in detail.

Author Response

I added 2 references showing that the continuum approximation used in this work is relevant even for very small nanoparticles (new references 4 and 5). There are indeed reports in the literature of the "crucial" role of "surface effects" in nano-objects. Unfortunately, at least for the eigenvibrations and material (gold) considered in this work, these reports are most often overstated and result from comparing experimental data to inappropriate model. See for example references 7 and 8 by Goupalov which show that an apparent reduction of the Young modulus in nanowires compared to the bulk material was in fact not true.

Round 2

Reviewer 3 Report

The manuscript can now be recommended for publication.